# Precision Landing Tests of Tethered Multicopter and VTOL UAV on Moving Landing Pad on a Lake

**DOI:** 10.3390/s23042016

**Published:** 2023-02-10

**Authors:** Cezary Kownacki, Leszek Ambroziak, Maciej Ciężkowski, Adam Wolniakowski, Sławomir Romaniuk, Arkadiusz Bożko, Daniel Ołdziej

**Affiliations:** 1Robotics and Mechatronics Department, Faculty of Mechanical Engineering, Bialystok University of Technology, Wiejska St. 45C, 15-351 Bialystok, Poland; 2Automatic Control and Robotics Department, Faculty of Electrical Engineering, Bialystok University of Technology, Wiejska St. 45C, 15-351 Bialystok, Poland

**Keywords:** tethered multicopter, VTOL UAV, precision landing, mobile landing pad, autonomous take-off and landing, position tracking

## Abstract

Autonomous take-off and landing on a moving landing pad are extraordinarily complex and challenging functionalities of modern UAVs, especially if they must be performed in windy environments. The article presents research focused on achieving such functionalities for two kinds of UAVs, i.e., a tethered multicopter and VTOL. Both vehicles are supported by a landing pad navigation station, which communicates with their ROS-based onboard computer. The computer integrates navigational data from the UAV and the landing pad navigational station through the utilization of an extended Kalman filter, which is a typical approach in such applications. The novelty of the presented system is extending navigational data with data from the ultra wide band (UWB) system, and this makes it possible to achieve a landing accuracy of about 1 m. In the research, landing tests were carried out in real conditions on a lake for both UAVs. In the tests, a special mobile landing pad was built and based on a barge. The results show that the expected accuracy of 1 m is indeed achieved, and both UAVs are ready to be tested in real conditions on a ferry.

## 1. Introduction

The increase in the importance of unmanned aerial vehicles in our lives is indisputable and obvious. Applications of unmanned systems are found in use over land but also over water reservoirs [1,2,3]. The latter applications are particularly interesting, because they often require the UAV (unmanned aerial vehicle) to start and end an air operation on a vessel [4]. This is especially important when UAV air operations take place far from the shoreline, e.g., in search and rescue operations or environmental or threat monitoring [5,6].

Most of today’s UAV systems are designed to be launched and to operate from a stationary point, therefore, they usually fail in applications (such as using a UAV during patrol missions of Coast Guard boats, military armored vehicles, or autonomous vessels for the purpose of increasing the range of field of view during monitoring of surroundings) where autonomous take-off and landing from moving landing pads are required. This also concerns their cooperation with autonomous mobile robots, which then become non-stationary landing pads for them in such applications [7,8]. Therefore, the problem of autonomous take-off and landing was also a leading issue in the project under the acronym AVAL (autonomous vessel with an air look), realized by the authors. This paper presents research results from that project, the main aim of which was to develop a control system for an autonomous vessel supported by UAVs to recognize dangerous objects in the water before a collision becomes unavoidable and damages the vessel’s body. Vessels cannot stop immediately due to their huge mass, thus, to achieve the ability for UAVs to monitor unexpected objects, they should take off and land on demand while the vessel is cruising at its full speed. This objective of the project is the main motivation for carrying out the presented research stage before the research will be moved to real conditions on the “Wolin” ferry. In the project, it was assumed that two kinds of UAVs (unmanned aerial vehicles) will be used, i.e., multicopter and VTOL(vertical take-off and landing) fixed-wing UAVs (Albatros). Both UAVs must operate from the deck of autonomous vessels, and their role is to detect obstacles ahead at a distance, ensuring a safe avoidance maneuver for the vessel. They can be further extended with various sensors acting for the benefit of man and the environment [9,10,11]. To realize this crucial functionality, a system composed of UAVs with built-in autopilot, an onboard computer, and a landing pad navigational station supported by UWB (ultra wide band) technology [12,13,14] was designed and built. Next, a series of experiments was planned to include tests in inland conditions [15], on a lake, and in real maritime conditions on a ferry [16,17]. This article will focus only on tests on the lake, 10 trials of take-offs and landings for each kind of UAV, and the primary aim is to achieve landing accuracy below 1 m at a landing pad speed of up to 5 km/h. The results obtained allow for advancing our research to the final stage, i.e., flight tests on the “Wolin” ferry during a cruise over the Baltic Sea [16]. The paper is composed of six sections. The first section is the introduction, where an analysis of literature related to the research is provided. The next section describes the designs of both the tethered multicopter and the fixed-wing vertical take-off and landing unmanned aerial vehicle (VTOL UAV—Albatros), followed by descriptions of the onboard computer, landing pad navigational station, and applied UWB system. In the third section, there is a presentation of the algorithms that were implemented to achieve the functionalities of tracking the position of the moving landing pad and flight phase management. The next section is devoted to the methodology of performed tests and how they were planned. In the fifth section, research results are presented and discussed. The article ends with conclusions.

### Current Advances in Related Research

The presented research fits into the trend of developing unmanned aerial systems for offshore applications. In successive years, many research teams directed their efforts to overcome different obstacles connected with the UAV’s operations in maritime conditions. The limited resources in terms of specialists, work hours, or free space available on the decks of the ships and boats are highly demanding and force the engineers to develop autonomous, reliable, and maintenance-free systems. However, creating fully autonomous solutions for commercial applications is extremely expensive and unaffordable for most companies and research facilities, thus advances are made gradually in narrow fields. One of the applications that utilities the UAV’s advantages is photogrammetry. It is widely used to create specialized or multipurpose maps of land areas for applications such as smart agriculture and farming [18] or for general purposes [19]. With the development of high quality sensors and the revolution in machine learning, photogrammetry may be used for bathymetric measurements in order to create precise depth maps of the ocean floor. In the research by Agrifitos et al. presented in [20], previously collected LIDAR data were used to train the SVM (support vector machine) model working on aerial photos taken by the UAV to determine a high-density model of the sea floor. The achieved results are promising at small depths. Due to the large scale of the potential mapping areas, high autonomy and usage of the mother vehicle moving on the water makes economic sense. In all kinds of aerial imaging tasks, data resolution is a crucial parameter. One of the solutions for overcoming small size limitations or environmental constraints is artificial enhancement of the acquired information. In the research presented in [21], the group of SR (super resolution) methods was discussed (for example, based on the generative adversarial networks (GAN) concept). Another field where UAVs are widely used are SAR (search and rescue) missions. VTOL platforms are frequently used for wide-area and monotonic marine surveillance. Many systems based on these platforms are designed for this task, Refs. [22,23] and many others take this application into account during the development process [24]. In addition to the advances in data acquisition and processing in particular applications, a UAV’s performance in the task universally depends on its parameters and ability to follow the system’s commands. In this context, precise localization is a crucial factor in almost any autonomous robot system. Although maritime conditions are characterized by the near-absence of obstacles blocking the signal from GNSS satellites, which are visible low over the horizon, the unpredictable ocean weather may disrupt satellite systems. Considering this, traditional navigation for drones becomes less reliable, especially when the optical flow systems are much less accurate over the water’s surface. In numerous UAV research applications, the assets of RTK (real time kinematic) systems are exploited. In the research by Lewicka [25], the significance of precise trajectory tracking with GNSS-RTK for collection and localization of the aerial images was highlighted. In [26], the utilization of GNSS-RTK for data fusion in geospatial applications was presented. In the seashore environment, precise localization was used for high quality spatial analysis incorporating terrestrial laser scanning, bathymetric survey, photogrammetry, and analogue archival bathymetric map data. Despite the fact that the RTK station on the moving landing pad may measure the global position with significant drift, correction of signal disruptions makes crucial relative positioning much more precise. In addition to GNSS systems for long range (global) navigation and vision systems for landing pad approaches, there are also many intermediate level systems, mostly utilizing radio frequencies. With different constraints and effectiveness, many commercial wireless data transfer systems (Wi-Fi, Bluetooth, etc.) are used for this purpose. A review of this topic was conducted by Yang and Yang in [27].

In the research conducted by our team, the two critical moments in the mission are take-off and landing. The dynamic characteristics of surface vehicles depend on their scale, type, and weather conditions. In the typical mission scenario, the vessel will perform translational movement along its axis, vertical movement caused by the waves, and its roll angle will oscillate. The system must be resistant to these disturbances. An analysis of the complexity of landing on moving and rocking platforms is presented in [28]. The solution incorporated not only the software and control part but also a mechanical adaptive leaning gear. A similar solution was presented by Tang et al. in [29], where an adjustable tripod was improved with omni wheels. In [30], a system for landing on moving vessels or vehicles was presented. Due to the goal of simplifying the UAV, all the aggregating calculations and data analysis were performed by the ground station on the landing pad. The relative position was obtained through sensor fusion by means of a stereo camera, fiducial markers, and DNN (deep neural networks). The predicted position was calculated by the Kalman filter supplied with the fused visual data and telemetry of IMU (internal measurement unit) signals. The system’s logic, based on a finite state machine, allowed for adaptation of the trajectory according to the tracking quality or the presence of the ground effect. The detailed review and analysis of the developed mobile landing platforms is a part of the publication [31] by Grlj et al. The aspects of take-off, landing, position estimation, control, trajectory generation, and docking systems were explained. In [32], the visual localization of the moving landing pad was implemented with robotic operating systems, which makes this approach similar to our solution. The very popular and increasingly frequently used vision systems are not the only way to indicate the relative position of the UAV in the USV local coordinate system (or vice versa). A set of ultrasonic sensors was mounted on the USV to detect the multirotor over the landing pad in [33]. The coupled navigation system for both robots was introduced with the newly developed guide point generation algorithm. In [34], the heterogenous mobile landing pad was equipped with a manipulator arm and docking terminal, matching the socket on the bottom of the UAV. The precise localization is based on fiducial markers. The applied controller fuses reactive, predictive, and optimal approaches. Due to the high degree of freedom of the robotic arms, the system is highly redundant. Achieved effective cooperation makes it possible to combine a wide field of view of the UAV and high payload capacity of the wheeled vehicle (possible charging station for the aerial robot), effectively exploiting their symbiotic cooperation. Other research on visual tracking of the landing pad that was actively referred to was published by Palafox et al. [35]. In [36], a tethered UAV was proposed; however, in contrast to the system presented in our research, it is mainly used not to supply power but to detect the relative position between aerial and ground units. The geometry of the loose tether is estimated by measuring the state of the mechanical components, which provides information about the robots’ positions in local coordinate systems. A more unusual approach was presented in [37], where a solar powered USV (unmanned surface vehicle) was introduced for recovery of the intentionally splashed-down waterproof UAV. However, the versatility of this concept does not apply to large-scale ocean vessels.

## 2. Architecture of UAV Systems

An overview of the presented system is depicted in Figure 1. As can be seen, two main parts can be distinguished: aerial—based on two types of UAVs for different applications, and water-based—mounted on the landing deck of the barge (vessel). The relations and connections between subsystems are annotated on the diagram, and their precise specifications are described in the sections that follow.

### 2.1. VTOL UAV

The VTOL type UAV is currently one of the most dynamically developing unmanned technologies [38,39,40]. The VTOL platform (Figure 2) used at this stage of research was a prototype on a 1:1 scale. It was meant to simulate the dynamics and attitudes of the final version of the aircraft; however, it was optimized for the purpose of extensive testing in field conditions. It also enabled the research team to elaborate procedures for missions in maritime conditions and verify compliance with safety regulations. The base for the airframe was an Applied Aeronautics Albatross fixed-wing platform equipped with one pushing motor, slightly forward-leaning wings, and two tail beams supporting the A-tail, which is sometimes called the ruddevator. The Albatross’s wings are equipped with two flaps and two ailerons. The whole structure (e.g. fuselage, wings, tail, etc.) is made of fiberglass and carbon fiber with wooden and carbon plates supporting the structure, which is a common technique [24]. This particular platform was chosen because of its parameters and flexibility for modifications. According to the manufacturer’s specifications, it is capable of taking off with a maximum weight of 10 kg, including a 4.4 kg payload. With a cruise speed of 19 m/s and max speed of 35 m/s, it can withstand a wind speed of 9 m/s. On the basis of our field tests in different configurations, the operational flight time was estimated between 1 to 4 hours depending on mission parameters. These high performance margins allowed for redesigning without compromises affecting the field trials. The Albatross was adapted to the VTOL configuration through the addition of easily removable polyamide wing inserts with attached carbon fiber beams (serving as arms for mounting four motors for MC (multicopter) functions). This enabled quick and easy conversion from the A-Tail fixed-wing aircraft to standard VTOL with multicopter functions.

Further modifications introduced a strengthened forward landing gear with a turning servomechanism and removal of the wheel, reinforcement of the underside of the fuselage with additional layers of carbon fiber fabric, and stiffening of flaps and removal of their servos. For safety purposes, floating inflatable buoys were attached to the landing gear and tail beams during tests without switching to fixed-wing mode (Figure 2, No. 6). Additionally, a safety tether mounting point was fastened underneath the fuselage for scenarios involving a tether. The propulsion for the fixed-wing mode was provided by a 3×16×10 Biela CFK composite propeller powered by a Turnigy SK3 Aerodyne 5055-430kV motor (Turnigy RC Power System, Kwun Tong, Hong Kong) which was the original drive included with the Albatross airframe (Figure 2, No. 5). This setup assured quick and smooth transition to FW (fixed-wing) mode. For the purposes of vertical take-off and landing, four T-motor MN5212 kV340 motors with G18×5.9 T-motor carbon fiber twin-blade propellers were mounted on the noted removable beams (Figure 2, Nos. 1–4). They allowed for a maximum total thrust of 17.42 kg. The battery pack used in the missions described in this paper was made of two SLS Quantum 25/50C 10Ah 6S Li-Po batteries connected in parallel, each weighing 1.16 kg. The total take-off mass was 9 kg. The main parameters of the modified VTOL UAV platform are presented in Table 1. The onboard equipment of the Albatross VTOL UAV includes a PX4 autopilot with custom firmware. The onboard control system enables automatic flight control of the VTOL UAV in different phases of flight (automatic take-off and landing in both MC (multicopter) and FW (fixed-wing) mode, switching from FW mode into MC mode and vice versa, landing pad, and waypoint following functions). Additionally, the VTOL UAV was equipped with an on-board computer communicating with the autopilot by wire connection. The onboard computer runs the implemented relative positioning algorithm needed for tracking of the mobile landing platform and precise positioning of the VTOL UAV. The VTOL UAV is able to communicate with the landing pad and air traffic control station by two separate radio links operating at 868MHz and 433MHz frequencies. The antennas for the communication links and for the GNSS (global navigation satellite system) were mounted outside the hull (Figure 2, Nos. 7–8 and 9 respectively). Nos. 10 in Figure 2 is a pitot tube.

### 2.2. Tethered Multicopter

The main unmanned aerial vehicle used in the AVAL project was a multirotor type helicopter. The UAV has a six-arm configuration with an individual propulsion unit at the end of each arm. Dynamic requirements in the control process implied that BLDC (brushless DC) electric motors had to be selected. The motors are stationed in a fixed manner relative to the arm. High-quality 17 × 5.8 diameter carbon propellers are attached to the rotating part of the motor. The motors are controlled by a dedicated electronic speed controller (ESC) necessary for these types of electric motors. Such a drive unit is able to generate a thrust of up to 40 N. All drives were tested and their characteristics were determined and mathematically modeled to maximize their potential [41,42,43]. In such a configuration, the total thrust of the multirotor is 240 N. With the helicopter’s net weight of 10.5 kg and the weight of the 50 m tether (power delivery cable) and optoelectric equipment (e.g., camera), we achieved a maximum take-off mass (MTOM) of 13.5 kg. The reserve of thrust in relation to the MTOM is sufficient and does not differ much from the desired ideal ratio of 2:1. This ensures high responsiveness of the system, necessary for precise maneuvers to stay above the landing pad—especially in the take-off and landing phases, at a relative wind speed as high as 60 km/h. The most important parameters of the multicopter are presented in Table 2. Significantly, for the developed multirotor system (Figure 3A), it is powered by tether from the ground/deck of the ship (Figure 3B). A special power cord (Figure 3C) supplies alternating current while the UAV is in flight. The automatic winch (Figure 3D) unwinds and collects the tether on its reel according to the UAV’s flight altitude. The UAV system implemented in this way allows for theoretically unlimited flight time (Table 2). In practice, of course, service stopovers or landings resulting from changes in weather or time of day should be provided for. The use of wired power supply makes this UAV independent from the flight time limitations resulting from battery capacity. A flying object of this class, i.e., with a mass greater than 10 kg, would have a useful flight time of 15 min. Under these assumptions, the above multirotor is a flying periscope extending the range of sight. Just climbing to a nominal height of 50 m above ground level and then descending to land in normal operation takes about 5 min. The remaining 10 min of battery use would be highly insufficient. A certain disadvantage of the presented flying platform is the high consumption of electricity. In atmospheric conditions with a wind speed of 4.5 km/h, i.e., very good weather, the energy demand is about 2000 W. The multicopter does not have airfoils that generate lift, so all its weight and the weight of the additional equipment has to be pulled upwards by the six rotors. On the other hand, we get very good controllability, the ability to operate in a tight space (very small helipad) with the surrounding equipment of an additional global and local navigation system, e.g., landing pad navigation station or UwB system tripods and masts. The multirotor control unit is an autopilot with dedicated firmware. It enables flight in manual mode (necessarily supported by the UAV automatic stabilization process) and fully automatic operation mode. The second mode of operation ensures automatic take-off at the set altitude, following of the moving landing pad and then precise landing without the participation of a human operator. Communication and viewing parameters are monitored throughout the flight via a telemetry link with an 868 MHz frequency. The transmission range and stability at such a short distance between the UAV-GCS is sufficient, but it can optionally be implemented through a modified power cable.

### 2.3. Landing Pad Navigational Station

In order to ensure precise UAV navigation in the neighborhood of the landing pad, the landing pad was equipped with a navigation station (LNS: Landing pad navigation station). This station measures the navigation parameters of the landing pad and exchanges data with the UAV’s flight computer via a radio link. The navigation parameters measured by the LNS are its current position in the WGS-84 (World Geodetic System 84’) system, speed, and course (heading). These parameters are sent to the UAV’s flight computer. In addition, the LNS station sends take-off and landing commands. The landing pad navigational station is shown in Figure 4. The mounted equipment on the landing pad can be seen in Figure 5 and Figure 6.

The station consists of a main computer (see ① in Figure 4), a GNSS module with antenna ②, an electronic magnetic compass ④, and a radio module ③ for LNS-UAV communication. For service/surveillance connection to the main computer, the station was equipped with a wi-fi router ⑤. The mechanical structure of the LNS was made of aluminum profiles.

The LNS measurement devices are connected to the LNS computing unit [44] located inside the main computer ①. An application running on the LNS computing unit collects all the station’s measurement data and manages the aircraft’s flight. Figure 7 shows all connections between LNS components.

The GNSS module [45] provides the landing pad position and speed, whereas the LNS course is measured via a calibrated electronic compass [46]. According to the manufacturer’s recommendation, the calibration process of the electronic compass involves recording 12 corrections of the magnetic deviation when the compass, along with the entire landing pad, makes a full rotation in the horizontal plane (12 corrections per full rotation give one correction datapoint every 30${}^{\circ }$).

### 2.4. Onboard Computer and Onboard System Structure

The onboard computer is an additional device that is used to run a relative positioning control algorithm. The onboard computer houses an ROS-based system (robot operating system) utilizing a Mavlink (micro aerial vehicle link) bridge to forward the messages transferred between the autopilot and the ground control station. As it is difficult to implement the control algorithm directly on the autopilot hardware, an Odroid XU4 computer was used.

The computer plays a key role in navigation, where it is most important to obtain accurate relative positioning data based on the UAV and the base station GNSS coordinates, and is further enhanced by including position information from the local positioning system (e.g., UWB). In order to provide this functionality, the onboard computer is also tasked with support and control of the UAV-mounted UWB tag.

The on-board computer connection diagram is shown in Figure 8.

The Odroid computer communicates with the Pixhawk autopilot using the UART protocol. The downstream connection is implemented over radio-link with the ground station using the serial protocol. An optional wi-fi connection allows for diagnostics and testing of overall system performance. The onboard computer communicates with the UWB tag over a USB connection.

The structure of the ROS-based onboard computer system is presented in Figure 9.

The ROS-based onboard computer system consists of a stack of the following nodes:*Communication_node*—This node is responsible for passing the Mavlink datastream from the ground station up to the autopilot and from the autopilot downstream to the ground station. The node retrieves the status and position information from the autopilot and publishes them in appropriate topics. Furthermore, the target position command and the GNSS position of the ground station are extracted from the upstream, then filtered and published. Finally, the node injects Mavlink commands issued by the commander node into the upstream link to facilitate the control of the position of the UAV relative to the landing pad.*Commander_node*—This program implements the relative position navigation algorithm and computes the Mavlink commands to be sent to the autopilot. The commands are calculated based on the desired target position and the estimated position of the UAV relative to the base.*Relative_position_node*—This node computes the position of the UAV relative to the base according to the GNSS sensor data. Because the GNSS sensor only provides coordinates and not the orientation of the frames, additional transformation is needed to extract the full pose.*Fusion_filter_node*—This node implements a Kalman filter to achieve the sensor fusion of the positioning information from the GNSS and the local positioning system (UWB).*UWB_node*—This node communicates with the onboard UWB system tag to provide position information from the local position system mounted on the moving platform.

### 2.5. UWB Positioning System

Ultra wide band technology was utilized to create a local positioning system, which acts as one of the elements in the relative positioning algorithm. The system is based on the lateration technique, where distance measurements are used in the process of position estimation. The ranging devices can be divided into two categories: anchors—stationary devices placed at the landing pad—and tag—ranging device mounted on the UAV. The distance measurements and the known positions of anchors are used in the process of estimating the tag’s position.

The special design of the positioning algorithm was implemented to fulfil specific requirements related to the navigation task of the fast-moving flying object and to integrate the positioning systems. The whole positioning algorithm’s design can be divided into three stages (Figure 10).

One of the most significant issues occurring in the aforementioned application is the possibility that the UAV will travel out of the range of the UWB measurement devices. In such a case, part or all of the distance measurements cannot be taken. There could also be a second issue related to the limited range of the measurement devices, occurring when the distance between the anchor and tag is near the maximum measurement range. When the tag is exposed to such conditions, intermittent distance measurement occurs, and measured values are strongly distorted. Total/partial blockage of the radio signal has a similar influence on distance measurements when a cable, tree, the UAV itself, a human, terrain, or any other obstacle is present in the signal’s propagation path. In order to withstand such complications, the positioning algorithm performs distance quality evaluation in the first step. At the end of the first stage, each distance measurement is categorized as: properly taken measurement, poor measurement, or unusable. The UWB modules applied in the positioning system send the update flag and the signal to noise (SNR) value through the data frame. These values are used to evaluate the quality of each measurement. In the first step, the update flag is checked. Each measurement has a dedicated bit in the update flag, where 0 means the measurement between a specific anchor and tag could not be taken. Next, the SNR parameter is considered. When a given distance measurement is taken but the ratio between the SNR and the distance drops below a given value, the measurement is considered as taken in poor conditions. If the SNR value is lower than a given threshold, the measurement is considered unusable. During the tests, the SNR threshold was set to −18 dBm. It is necessary to note that when the SNR value drops below −20 dBm, the measurements are not taken continuously, and their values happen to be tremendously inaccurate [47,48]. The threshold was set 2 dBm higher than the border at which such a phenomenon was observed. The threshold of −18 dBm makes it possible to achieve positioning ranges higher than 35 m and prevents dangerous error spikes in the positioning results, which can lead to unstable flight.

The evaluated distance measurements are then utilized during the second stage as the input of the extended Kalman filter, which can adapt to a different number of inputs in subsequent steps. The minimum number of usable distance measurements required to perform reliable estimation of the tag’s coordinates is four.

At the final stage, the parameter evaluating the achieved accuracy of position estimation is calculated. During this process, different factors are considered: number of correct measurements, quality of measurements, and distance to the tag. The UWB positioning system outputs the estimated relative position of the UAV together with the evaluated accuracy into the relative position integration algorithm. Such an approach makes it possible to neglect UWB measurements during the integration stage, when the UWB system cannot estimate the UAV’s position.

## 3. Algorithm of Take-Off, Following, and Precision Landing on a Moving Landing Pad

Performing take-off, following, and precision landing on a moving landing pad requires accurate information about the UAV’s (x,y,z) position relative to the landing pad. An additional piece of information necessary for precise UAV control is the velocity of the landing pad relative to the ground and its course. The last two parameters are estimated via the Kalman filter implemented on the UAV onboard computer—the measurement data (heading/course from the compass and velocity from the GNSS module) are supplied directly from the landing pad navigation station via the radio link with a frequency of about 2 Hz. The relative position of the UAV is also estimated on the UAV onboard computer and combines position measurements from GNSS modules (the LNS module and UAV module) and from the UWB system.

A flowchart of the take-off, following, and precision landing algorithm is shown in Figure 11. An application running on the supervisory computer (LNS computing unit in Figure 7) measures the position and velocity of the landing pad at a frequency of 2 Hz. These data, along with the UAV’s desired position, are sent to the onboard computer, also at 2 Hz. When the user issues the start command, the supervisory computer sends the “ARM” command to the UAV’s autopilot. It should be noted that all commands from the supervisory computer are sent to the drone’s onboard computer, which sends these commands further to the autopilot (see connection diagram shown in Figure 8). When the autopilot confirms arming, the supervisory computer sends a “TAKE-OFF” command to the drone’s autopilot and waits for its confirmation. When the take-off begins, the computer sends a command to the autopilot to enter the “FOLLOW-ME” mode, in which the drone will follow the landing pad according to the desired position. When the user issues a landing command, the supervisory computer changes the drone’s desired altitude to 2 m below the landing pad. The UAV continues in “FOLLOW-ME” mode, decreasing its altitude. At one point, the drone gently hits the landing pad, which the autopilot interprets as a landing. The UAV sends a confirmation landing message, which completes the whole process.

The standard controller implemented in the PX4 autopilot firmware was used to control the UAV’s altitude and heading. Controlling the UAV’s horizontal attitude required modifications of the standard autopilot algorithm. The task of the modified regulator was to control the horizontal velocity of the UAV so that it follows the moving landing pad. If the UAV’s desired horizontal velocity is denoted as ${\vec{V}}_{U_{D}}$ and the landing pad’s actual horizontal velocity is denoted as ${\vec{V}}_{L_{A}}$, the control law takes the form:(1)$${\vec{V}}_{U_{D}}={\vec{V}}_{L_{A}}+K_{P}\Delta \vec{r_{H}}+K_{I}\int_{0}^{T}\Delta \vec{r_{H}}dt$$
where $\Delta \vec{r_{H}}$ is the UAV–landing pad position error in the horizontal plane, and $K_{P},K_{I}$ are regulator gains. The complexity of the mathematical model of the controlled object necessitated manual tuning of controller gains during experiments.

Figure 12 shows the modified control architecture of the PX4 autopilot, where the modifications are marked in red and blue. The main difference from the original architecture is that the UAV’s desired horizontal velocity is directly applied to the “Velocity Control” block.

## 4. Tests on Mobile Landing Pad Moving on a Lake

The scope of the tests performed encompassed testing of the procedure for vertical take-off and verification of precision landing on the barge deck where the landing pad with a size of 6 × 6 m is located. The barge’s speed is limited to 5 km/h due to the pushing boat’s maximum speed and the lake’s size. The tests were carried out with the use of an airframe with vertical take-off and landing functions. The Albatross (VTOL UAV) test platform prepared for this test stage was equipped with additional inflatable floats, preventing the platform from sinking in the event of uncontrolled launching. The test program concerned the execution of flights (10 trials) consisting of the following phases:Automatic vertical take-off in MC mode from the barge’s deck once it achieved its maximum speed,Following the moving barge in MC mode at 4 m (Albatross) and 8 m (tethered multicopter) above the moving deck (AGL altitude),Automatic vertical landing on the barge’s deck while it was still in motion.

The control of individual flight phases and switching between them was performed by an external computer with radio communication with the on-board computer of the unmanned aircraft and the autopilot. The same method of flight phase control was applied to the tethered multicopter, as the overall system architecture is common for both UAV types. Before each trial, the Albatross was placed at the center of the landing pad. After the barge achieved its maximum speed, take-off was ordered. The VTOL UAV flew for a few minutes, tracking the barge at an altitude of 4 m above the landing pad’s surface, and then landed, and after that, the distance from the landing pad’s center was measured. In the tests, flight altitude was controlled in the following manner: during the take-off phase, the desired flight altitude was set automatically to the value being the predefined take-off altitude in the autopilot settings (required by the applied autopilot—PX4), i.e., 20 m, to achieve fast ascension, and just after the take-off command is issued, the landing pad navigational station overrides this value with the desired flight altitude calculated as the sum of GNSS altitude of the landing pad and desired flight altitude above landing pad level (4 m for Albatross and 8 m for tethered multicopter). During the landing phase and simultaneous tracking of the landing pad’s position, the landing pad navigational station sets the desired altitude of 2 m below the level of the landing pad to ensure achievement of touchdown, which can be seen on altitude plots as a negative altitude. After touchdown, the autopilot was disarmed automatically. The following video presents an example test performed during VTOL trials on a lake—https://youtu.be/MwTTg3yhhtQ (accessed on 1 January 2023). The same test procedure was repeated for the tethered multicopter, and this time, we performed 10 trials, but at a higher altitude of 8 m. The conditions of the tests were similar. A circle with a radius of 1 m was marked on the deck to make it possible to assess the landing precision more easily at a glance.

## 5. Results

### 5.1. VTOL Results

Out of the two studied UAV platforms, the VTOL was the one that was more exposed to the hostility of weather conditions (wind gusts, air density) due to its characteristics (lower thrust/weight ratio, additional wing area, lower take-off mass, and higher side wind vulnerability) in comparison to the more compact multirotor. The effects of this imbalance can be noted on the pitch and roll angle charts in Figure 13. Although the desired pitch angle value follows the measurement tightly (filtered by the internal autopilot Kalman filter), the desired roll angle and measured values diverge over time (especially within the timespan from 992 to 1005 s). Fluctuations of the pitch angle are negligible, whereas in roll, they are clearly visible in both desired and measured values. The regulation error in roll angle oscillates by around 3 degrees, with the uppermost value being around 5 degrees. The difference between the behavior of the VTOL and control of these two axes is explained by the noted disadvantages. When side wind pushes the plane to its left, the controller tries to maintain position and leans it to the right (increased roll angle). However, the force generated by the wind is not constant and varies due to imperfect symmetry of the fuselage, wing area exposed to the wind, and the gusts of wind themselves. These effects are more influential at higher altitudes, when the VTOL is above the treeline on the bank, which in this case is around 8 m. Correction of this error is also more challenging due to the higher moment of inertia around the longitudinal axis (wingspan is greater than the length of the plane, and the wings are heavier than the carbon tail rods).

The lower control quality with regard to roll angle affected the lateral movement velocity Vy.

Figure 14 shows that movement along the X axis within the body frame is realized most of the time with proper velocity setpoint tracking, characterized only by deviations that are few and far between. In contrast, tracking of Vy presents much lower quality and requires a longer time to cancel out the initial oscillations (caused by rapid take-off with control mode transition to custom ’follow me’). The relations, pitch angle—Vx and roll angle—Vy are not direct; however, they have a dominant influence on the horizontal acceleration and speed over the wind factor and turbulent airflow (assuming flight altitude is over the ground effect zone). This ultimately ends with divergent position drift in the forward/backward and sideways directions (Figure 15). The yaw angle graph is characterized by regular disturbances. All of them start with twisting of the VTOL nose in the same direction (to the right in Figure 13). They are most likely caused by a side wind pushing the plane’s tail. The yaw angle setpoint drift over time is caused by rotation of the landing pad, which was traversing the lake along a curved path. The altitude during the flight test is shown in Figure 16. The desired value was generated by shifting the measured landing pad position by a constant value (altitude setpoint, which in this case was 4 m). The instability of the desired value is the effect of the barge’s flotation and GNSS inaccuracy (this altitude is not filtered with IMU (inertial measurement unit) sensor fusion). The fact that initial and final values are lower than zero is caused by the fact that the GNSS receiver on the barge is mounted higher (around 50 cm) than the deck, and the VTOL’s altitude is measured relative to it. The actual value of the fused altitude might also be slightly affected by air pressure drift over time. As can be seen on the log timeline on the charts, take-off occurred over fifteen minutes (980 s) prior to autopilot initialization. There are also two visible effects on the setpoint chart. The first of them is the 20 m peak after take-off. It is the result of beginning the flight using the built-in take-off Mavlink command, which sends the vehicle to the default height. In our implementation, the flight mode is changed immediately after that, and the setpoint is then generated by our system. The second unusual effect is that the setpoint is −2 m during the landing procedure. This is the consequence of landing in follow-me mode. This way, the plane was prevented from hovering over the deck. The flight conditions are presented in Table 3.

Table 4 presents the results of landing precision verification for the VTOL UAV based on ten trials. The average landing distance from the center of the landing pad, its maximum and minimum values, and standard deviation are given. Figure 17 presents pictures of the VTOL UAV after landing for the best and the worst cases, i.e., for maximum and minimum distance. The average distance from the center of the landing pad after landing was 43.8 cm with a standard deviation of about 11 cm. This means that the required landing accuracy at 1 m was completely achieved.

Examples of logged VTOL flight parameters are shown in the Figure 13, Figure 14, Figure 15 and Figure 16.

### 5.2. Tethered Multicopter Results

Table 4 also presents the results of landing precision verification for the tethered multicopter based on ten trials. As previously, the average landing distance from the center of the landing pad, its maximum and minimum values, and standard deviation are given. Figure 18 presents pictures of the tethered multicopter after landing for the best and worst cases, i.e., for maximum and minimum distance from the center of the circle on the deck. The average distance from the center of the landing pad after landing was 38.95 cm with a standard deviation of about 22.04 cm. This means that the results for the tethered multicopter are also within the assumed range of 1 m, even if the flight altitude was higher than in the case of the Albatross and despite the influence of the tether. The tether, which also plays the role of power supply wire, applies additional force to the multicopter frame because of its weight and aerodynamic drag. The most difficult phase of the multicopter flight was landing and next a touchdown. Oscillations of orientation angles which appear just before touchdown are presented in Figure 19. Linear velocities given in the multicopter body coordinate frame are in Figure 20. In turn an exemplary plot of flight altitude is presented in Figure 21. The last figure i.e., Figure 22 shows an example of a path of flight in landing pad tracking mode achieved by multicopter during tests.

## 6. Conclusions

The results of the experiments presented in the paper were just preliminary tests prior to flights in real maritime conditions [16]. As a preparatory stage for an extremely difficult experiment, the results can be treated as significant proof of the operational state of both UAV systems. This particularly concerns the VTOL UAV—Albatross. The observed series of autonomous take-offs and landings guarantee repeatability of precision landing on the moving landing pad located on the barge. For the VTOL UAV, accuracy was within the range of 1 meter (exactly 43.8 cm with a standard deviation of about 11 cm), and for the tethered multicopter, results were even better (40.20 cm with a standard deviation of about 9.0 cm), noting that the multicopter was tethered and its flight altitude was higher. During flights, the average wind speed was about 3.1 m/s, and the barge speed was about 1.38 m/s (5 km/h). Thus, maximum relative speed (airspeed) was about 4.5 m/s. The results, especially plots of orientation angles (roll and pitch angle), indicate that the VTOL UAV is more sensitive to wind gusts. This property of the VTOL UAV should be paid due attention in the test on the sea. On the other hand, the roll and pitch angle plot for the multicopter shows that the most difficult part of the flight was the touchdown, when the multicopter’s orientation was unstable. Flight altitudes for the VTOL UAV and the multicopter were, respectively, 4 and 8 m, thus the climb and descending phases of the flight were relatively short. Tracking of the landing pad was active in a limited time window, but the speed of the barge was also limited to 5 km/h. The achieved precision of landing back on the landing pad at a value of 1 m and at a low speed of 5 km/h in the discussed tests should not be increased meaningfully in the next stage of experiments in inland [15] and in maritime conditions on a ferry [16], whose cruise speed is about 20 km/h. The results of this further research will confirm this conclusion clearly. The experiments on the lake can also be treated as tests in semi-real conditions, because they take into consideration the influence of environmental conditions such as the large surface of the surrounding water and wind on the course of flight. Therefore, these conditions differ from inland conditions and are more like those over the sea. During tests, it happened that each UAV almost touched the surface of the water a few times. The tethered multicopter even went underwater once due to unexpected behavior of the position tracking algorithms, which incorrectly calculated velocity vectors processed by the autopilot. Due to the experience gained from these tests, it was possible to correct all bugs in the code and tune the Kalman filter to avoid these kinds of problems during flights over the sea.

During the presented tests and the entire AVAL project, we found that tests in real-life conditions should be preceded by careful step-by-step investigations of all subsystems, and this is particularly important for successful research. This requires the involvement of specialists from different areas such as computer science, engineering science, aerospace, mechatronics, and robotics. In addition, fluent communication and cooperation among the entire team is crucial, because the licensed operator must control each flight with RC remote control, and, simultaneously, UAVs must be monitored by the UAVs’ ground station and landing pad navigational station operators. Any unexpected UAV behaviors are tracked from the landing pad navigational station, the UAV’s onboard computer, and the PX4 autopilot unit. Each test was carefully planned with an analysis of all potential dangerous scenarios, and before take-off, a detailed checklist was verified. Before the presented tests, many other trials were run in inland conditions, and many of them ended with UAV crashes.

In conclusion, it is particularly important to carefully prepare for experiments with UAVs in difficult environmental and weather conditions. Performing tedious repeatable exercises plays a crucial role in success in this area of research.

## Figures and Tables

**Figure 1 sensors-23-02016-f001:**
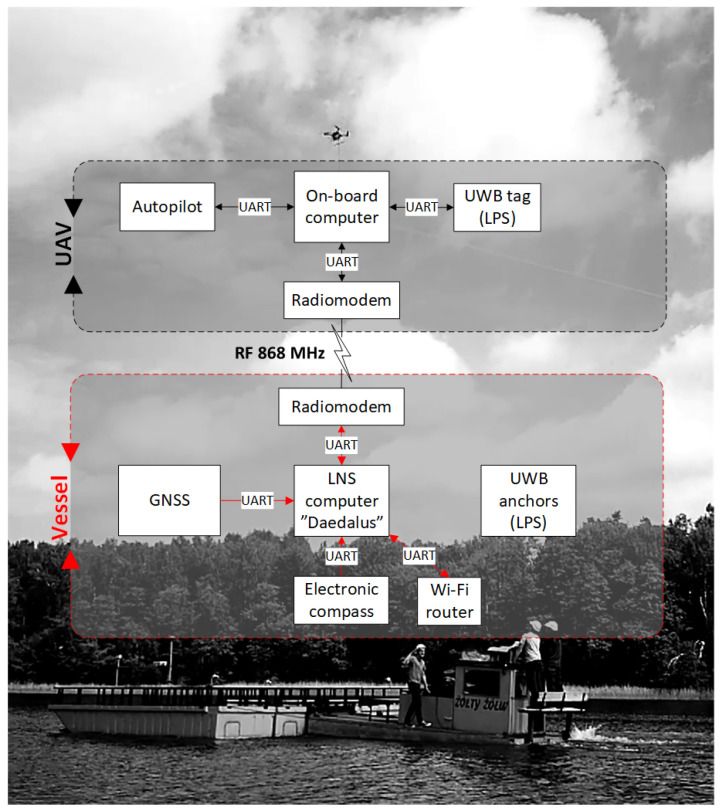
Components of the precision navigation system.

**Figure 2 sensors-23-02016-f002:**
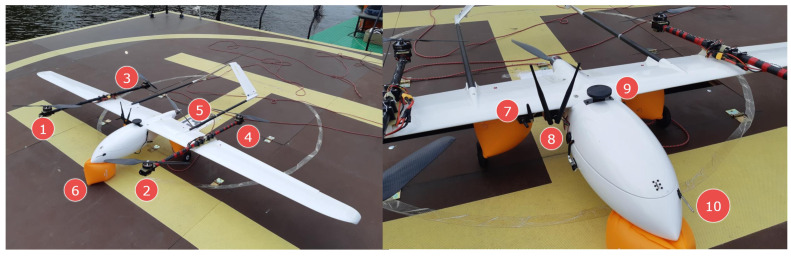
The Albatros VTOL UAV, 1,2,3,4—MC mode motors, 5—a FW pusher motor, 6—a inflatable buoy, 7—a 868MHz radiomodem, 8—a 2.4GHz radiomodem, 9—a GNSS antenna, 10—a pitot tube.

**Figure 3 sensors-23-02016-f003:**
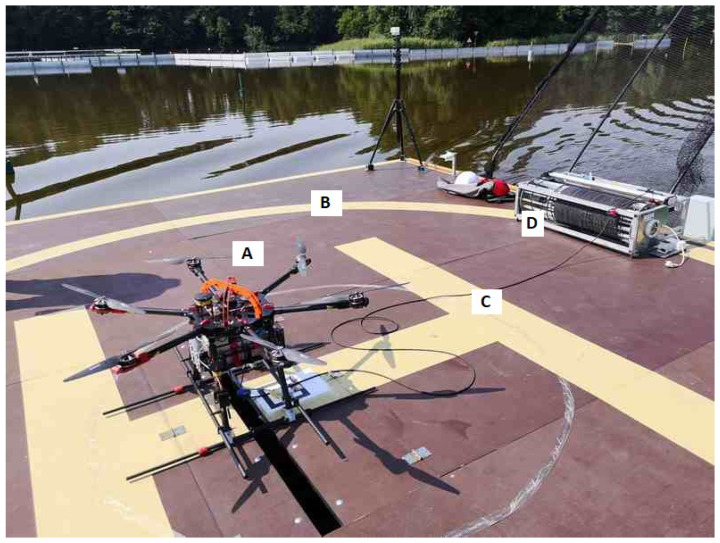
Mutlirotor UAV (A), landing pad (B), power cord (C) and automatic winch (D).

**Figure 4 sensors-23-02016-f004:**
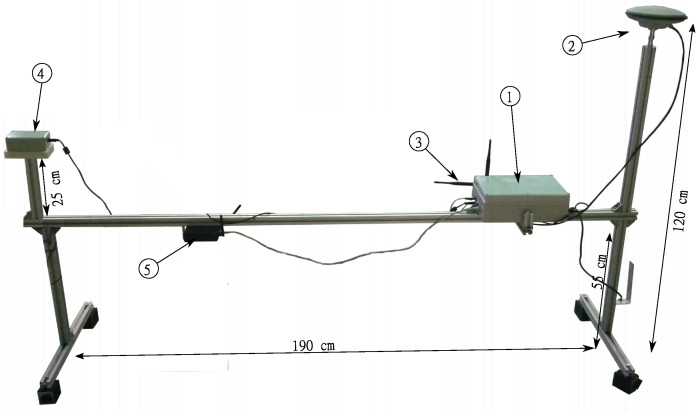
Landing pad navigation station, ①—a main computer, ②—a GNSS module with antenna, ③—radiomodems, ④—electronic compass, ⑤—a WiFi router.

**Figure 5 sensors-23-02016-f005:**
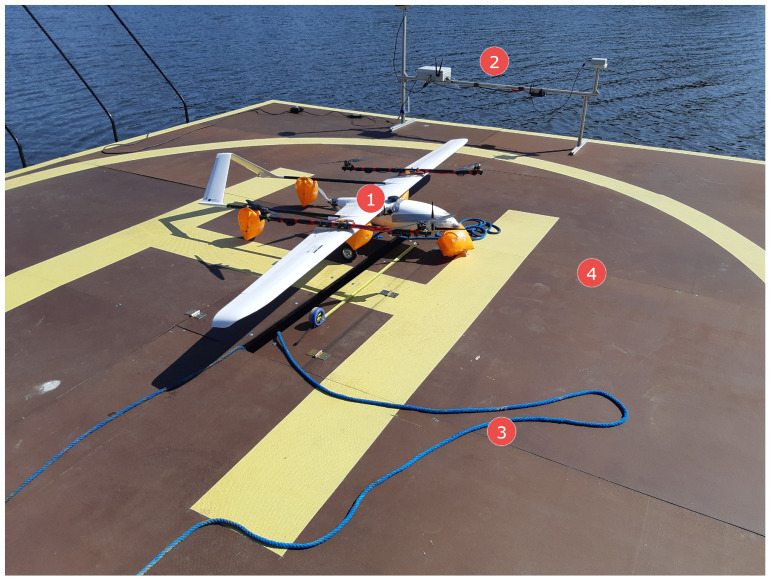
Albatross VTOL on a floating landing pad during lake tests, 1—Albatros VTOL UAV, 2—the landing pad navigational station, 3—a safety rope, 4—the landing pad.

**Figure 6 sensors-23-02016-f006:**
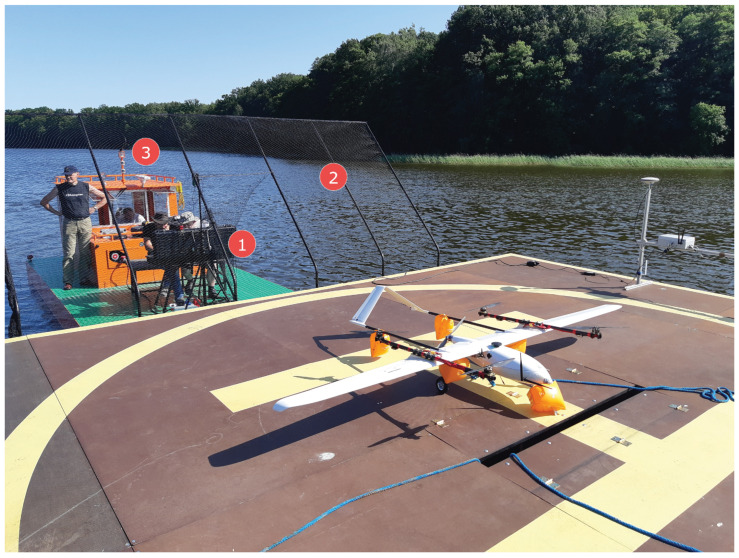
Albatross VTOL and floating landing pad view, 1—UAVs’ ground control station, 2—a safety net, 3—a push barge.

**Figure 7 sensors-23-02016-f007:**
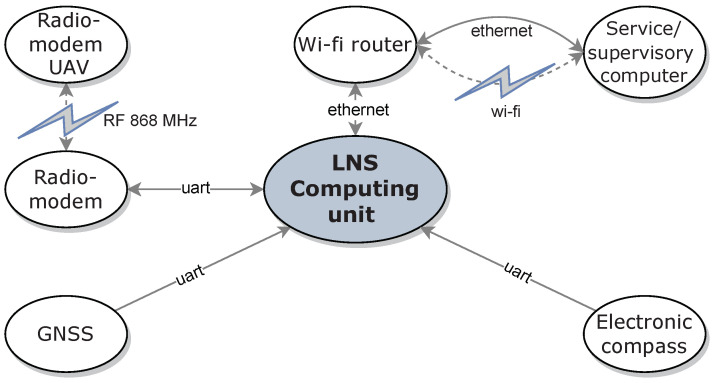
Diagram of LNS connections.

**Figure 8 sensors-23-02016-f008:**
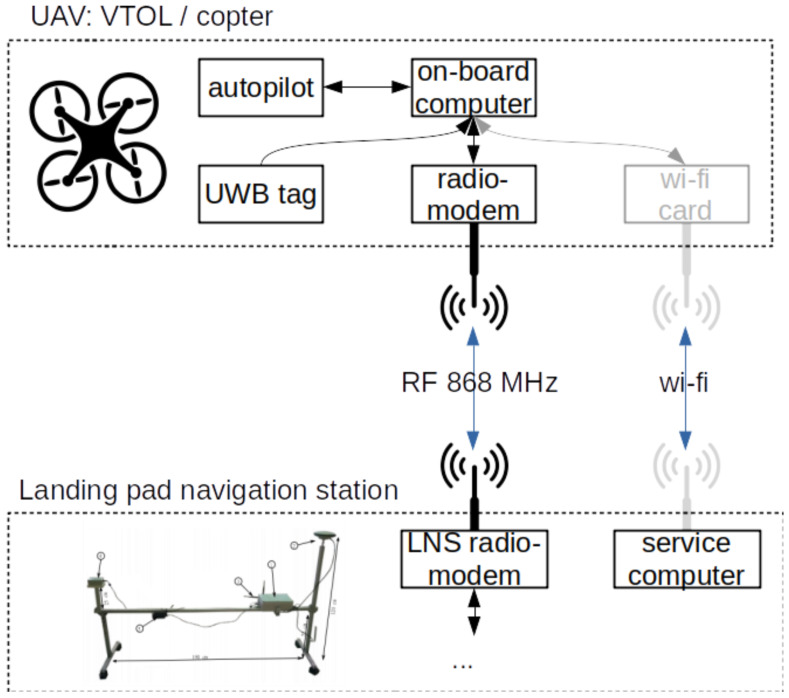
Diagram of onboard system connections.

**Figure 9 sensors-23-02016-f009:**
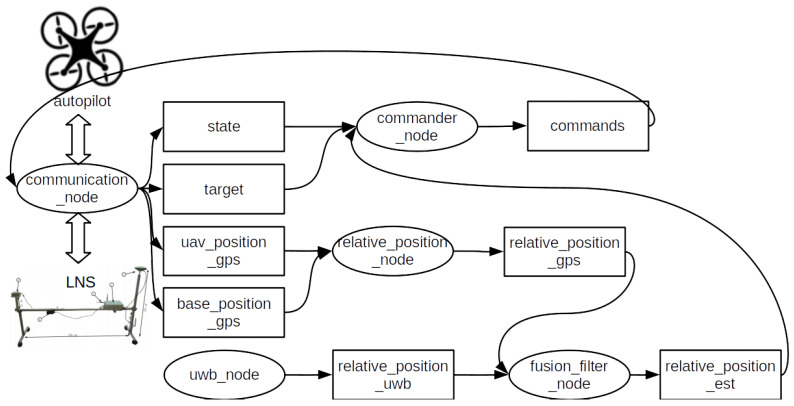
Diagram of the ROS-based control system for the onboard computer. ROS (robot operating system) nodes are shown as ovals and ROS topics are presented in rectangles.

**Figure 10 sensors-23-02016-f010:**
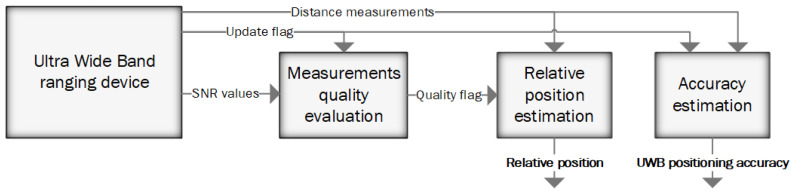
Diagram of the UWB positioning algorithm.

**Figure 11 sensors-23-02016-f011:**
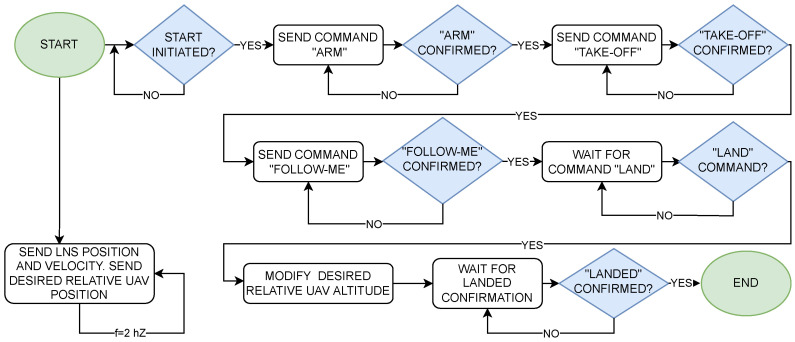
Algorithm flowchart.

**Figure 12 sensors-23-02016-f012:**
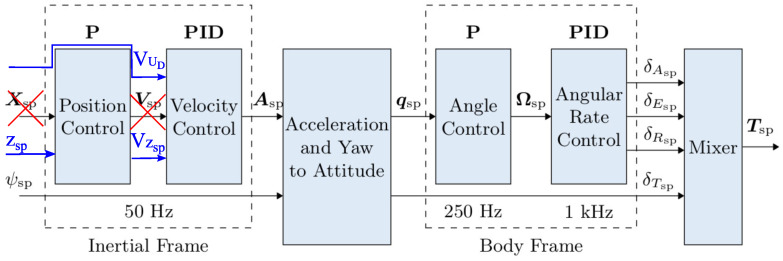
Modified autopilot control architecture [49].

**Figure 13 sensors-23-02016-f013:**
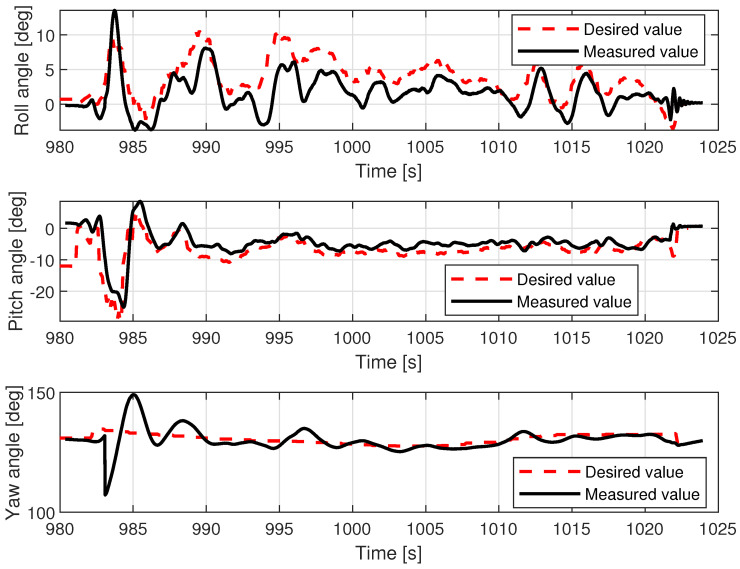
VTOL roll, pitch, and yaw angles during the in-flight test.

**Figure 14 sensors-23-02016-f014:**
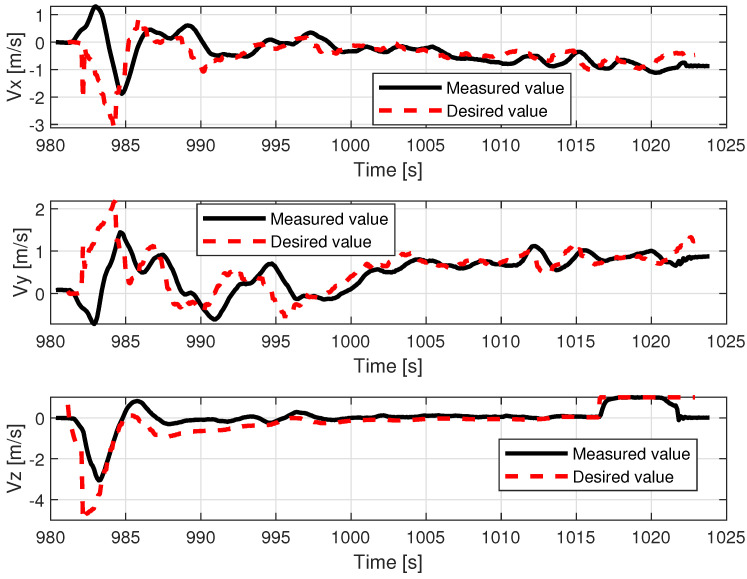
VTOL linear speeds during the in-flight test.

**Figure 15 sensors-23-02016-f015:**
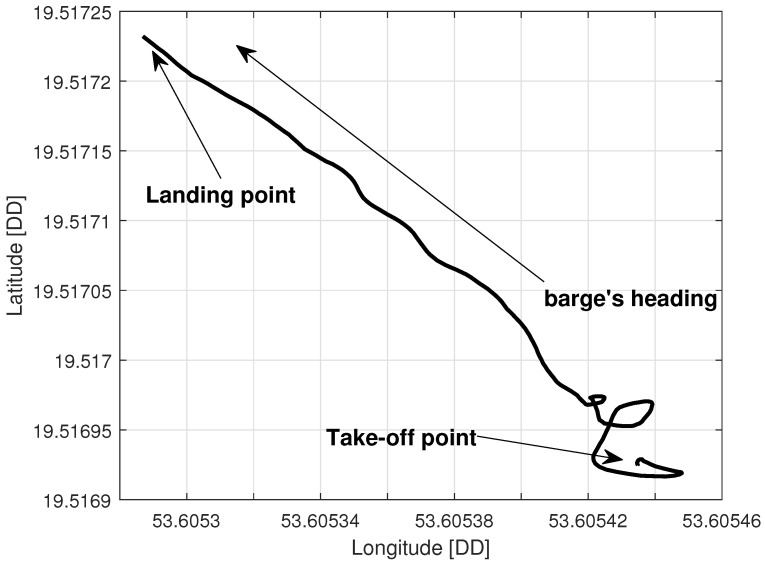
VTOL path during the in-flight test.

**Figure 16 sensors-23-02016-f016:**
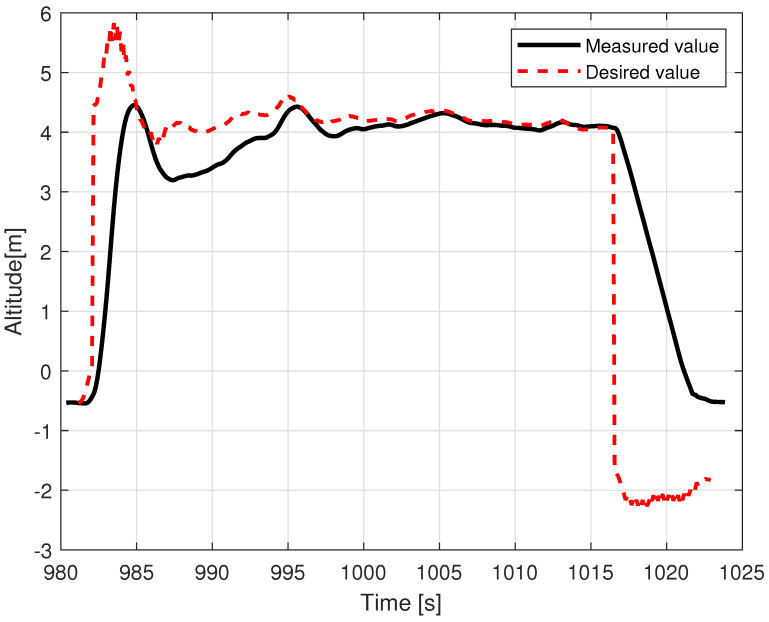
VTOL altitude during the in-flight test.

**Figure 17 sensors-23-02016-f017:**
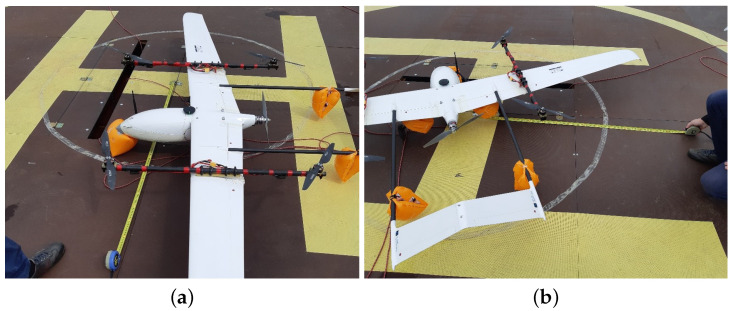
Albatross VTOL: (**a**) worst result—landing at a distance of 65 cm from the center of the landing pad; (**b**) best result—landing at a distance of 26 cm from the center of the landing pad.

**Figure 18 sensors-23-02016-f018:**
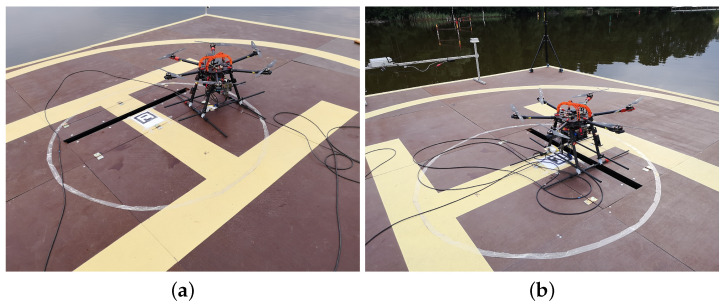
Tethered multicopter landing: (**a**) worst result—landing at a distance of 61 cm from the center of the landing pad; (**b**) best result—landing at a distance of 31 cm from the center of the landing pad.

**Figure 19 sensors-23-02016-f019:**
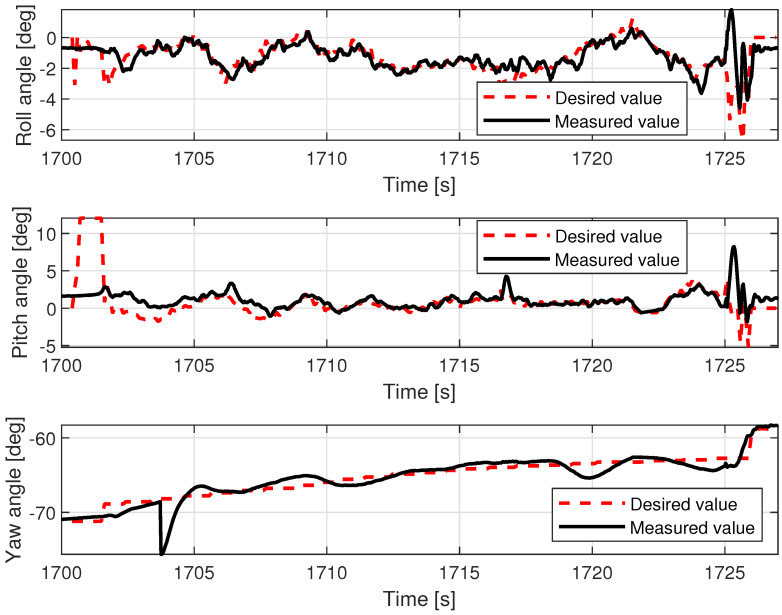
Multicopter roll, pitch, and yaw angles during the in-flight test.

**Figure 20 sensors-23-02016-f020:**
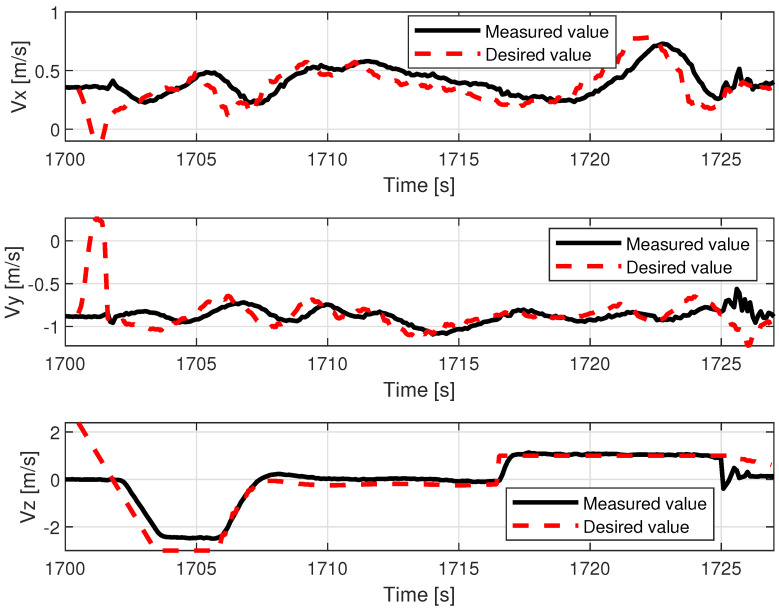
Multicopter linear speeds during the in-flight test.

**Figure 21 sensors-23-02016-f021:**
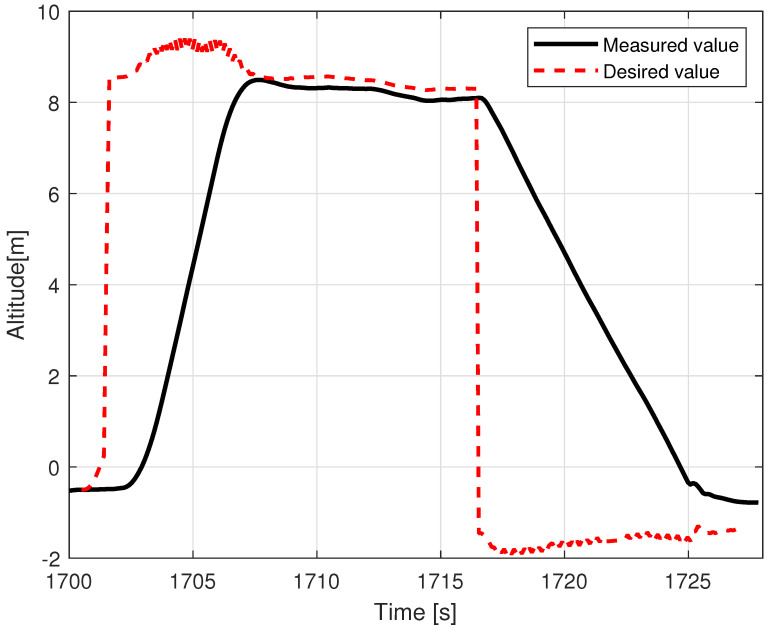
Multicopter altitude during the in-flight test.

**Figure 22 sensors-23-02016-f022:**
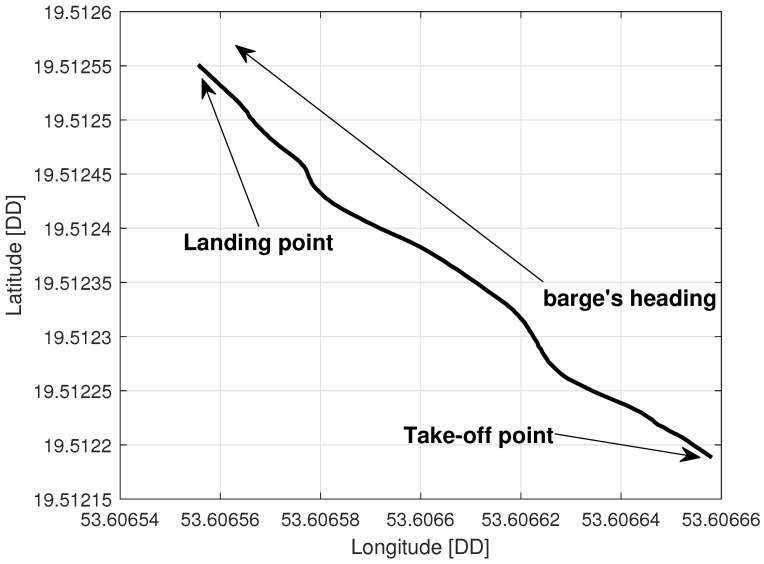
Multicopter path during the in-flight test.

**Table 1 sensors-23-02016-t001:** Main VTOL characteristics.

Parameter	Value [Unit]
Wingspan	3.1 [m]
MTOW (Maximum Take Off Weight)	10 [kg]
Battery	2 × 10,000 [mAh], 6S, 22.20 [V]
Motor Voltage	44.40 [V]
Battery capacity	12 [Ah]
Flight duration MC	15 * [min]

* In the default configuration for the purposes of the tests.

**Table 2 sensors-23-02016-t002:** Multicopter parameters.

Parameter	Value [Unit]
Dimension between opposite motor axles	960 [mm]
Max.wheelbase diameter	1400 [mm]
Maximum total drive thrust	240 [N]
MTOW	13.5 [kg]
Payload	3 [kg]
Flight altitude	Up to 50 [m]
Power type	Electric, tethered from ground
Motor voltage	22.2 [V]
Wind resistance	60 [km/h]
Construction	Modular - equipment can be adapted to needs
Flight mode	Semi-manual or automatic
Flight duration	Unlimited *

* Theoretically limited only by mechanical state of the components.

**Table 3 sensors-23-02016-t003:** Weather conditions during the tests.

Parameter	Value [Unit]
Average wind speed	3.1 [ m/s ]
Average air temperature	22.1 [°]
Average air humidity	73.2 [%]

**Table 4 sensors-23-02016-t004:** Results of VTOL UAV and tethered multicopter landing tests.

UAV Type	Number of Trials	Mean Dist.L	Std. Dev.	Max	Min
VTOL UAV	10	43.8 [cm]	11 [cm]	65 [cm]	26 [cm]
Multicopter	10	40.20 [cm]	9.0 [cm]	61 [cm]	31 [cm]

## Data Availability

Not applicable.

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
