# Peer review of "Precision Landing Tests of Tethered Multicopter and VTOL UAV on Moving Landing Pad on a Lake"

_sensors, 2023, doi:10.3390/s23042016_

Round 1
Reviewer 1 Report
I send the review in the attachment.

Author Response
Dear Reviewer!
Thank you for your valuable comments. Our answers to your specific questions are in the attached file.
Best regards
Cezary Kownacki

Reviewer 2 Report
1. Need to define the full form of UAV, VTOL, ROS, UWB
2. Insufficient introduction. Write briefly about the need, motivation, and scope.
3. Insufficient literature and a Separate chapter for literature survey with relevant research paper required. (Latest by 5 years like 10.3390/drones6010017, https://doi.org/10.3390/s22010404 , https://doi.org/10.3390/app11199157, 10.1007/978-3-319-70836-2_23 etc and many more)
4. line 23. Mention the applications clearly.
5. Line 33: Onboard computer or onboard controller?
6. Line 66: Why material for the structure is fibreglass and carbon fibre with wooden and carbon plates? Need extensive literature to support this statement. As the material selection is equally important.
7. line 71: the basis of estimation of operational time?
8. Line 75: MC function?
9. Add one more sub-section under the 2.1 section as design and development of VTOL UAV?
10: line 86: FW mode
11. Line 103: GNSS system?
12. Table 1: MTOW or MTOWM?
13 Sections 2,3 and. should be subsections of 2.2?
14 Line 156: WGS-84 system specification? Details?
15. Line 174 Why 30 deg?
16. Line 181: Does the computer plays a key role in navigation? Can you justify it?
17. Line 178/194/199//203//362: MAVLink means Micro Air Vehicle Link?
18: Figure 7: links are Not visible. Redraw.
19 what is the measured SNR Value for your proposed system? Compare that one with the literature.
20. How did you decide on SNR -18DdBm?
21. Line 268-269: How do you calculate desired horizontal velocity?
22. Section 3: Draw a flow chart for the algorithm. Need explanation with proper steps which are followed in the algorithm.
23. Line 308: 2m desired altitude is too much compared to the dimension of the copter.
24. Line 311: How can you put a video link in the text? Put stats only. Prepare a table for all specifications of designed parameters.
25. Line 317/ 320: Comparison table of two methods required. Show similarities also.
26: Combine tables 3 and 4
27: What is meant by dominant influence. Justify?
28: Line 355: IMU?
29. Line 369,370,371: Prepare table. Write units. . Write the impact of these parameters on both systems
30: Line 397: 38,95 or 38.95 cm and 22,04 cm or 22.04 cm?
31: line 402: 1 m / one meter? maintain uniformity
32. Author has shared experience but not a firm conclusion from the result and discussion section.
33.. Write a conclusion for a better system and things not to do while designing the system.
The first letter of abbreviations should be capitalised when they appear the first time in the text.
need to add mathematical support for the results.
Author Response

(The authors gave the same response as above.)

Reviewer 3 Report
The authors of the article have done an extensive and appreciable experiment on the said topic.
The article is good in its current state however can be further improvised by the following points in my opinion
1. A mathematical model to support the experiment can be portrayed
2. Simulational results can be used to check in other environments than a lake.
3. Results can further be used to benchmark with other similar approaches
Author Response

(The authors gave the same response as above.)

Round 2
Reviewer 2 Report
Accept in current form.